# The Management of Obstructive Sleep Apnea Patients during the COVID-19 Pandemic as a Public Health Problem—Interactions with Sleep Efficacy and Mental Health

**DOI:** 10.3390/ijerph20054313

**Published:** 2023-02-28

**Authors:** Anca Diana Maierean, Damiana Maria Vulturar, Ioana Maria Chetan, Carmen-Bianca Crivii, Cornelia Bala, Stefan Cristian Vesa, Doina Adina Todea

**Affiliations:** 1Department of Pneumology, Iuliu Hatieganu University of Medicine and Pharmacy, 400332 Cluj-Napoca, Romania; 2Morphological Sciences Department, Iuliu Hațieganu University of Medicine and Pharmacy, 400000 Cluj-Napoca, Romania; 3Department of Diabetes and Nutrition, Iuliu Hațieganu University of Medicine and Pharmacy, 400332 Cluj-Napoca, Romania; 4Department of Pharmacology, Toxicology and Clinical Pharmacology, Iuliu Hatieganu University of Medicine and Pharmacy, 400337 Cluj-Napoca, Romania

**Keywords:** SARS-CoV-2, obstructive sleep apnea, diagnosis devices, mental health, therapies

## Abstract

With the onset of the COVID-19 outbreak, it was stipulated that patients with obstructive sleep apnea (OSA) may have a greater risk of morbidity and mortality and may even experience changes in their mental health. The aim of the current study is to evaluate how patients managed their disease (sleep apnea) during the COVID-19 pandemic, to determine if continuous positive airway pressure (CPAP) usage changed after the beginning of the pandemic, to compare the stress level with the baseline, and to observe if any modifications are related to their individual characteristics. The present studies highlight the level of anxiety, which was high among patients with OSA during the COVID-19 pandemic (*p* < 0.05), with its influence on weight control (62.5% of patients with high levels of stress gained weight) and sleep schedule (82.6% reported a change in sleep schedule). Patients with severe OSA and high levels of stress increased their CPAP usage (354.5 min/night vs. 399.5 min/night during the pandemic, *p* < 0.05). To conclude, in OSA patients, the presence of the pandemic led to a greater level of anxiety, changes in sleep schedule and weight gain because of job loss, isolation, and emotional changes, influencing mental health. A possible solution, telemedicine, could become a cornerstone in the management of these patients.

## 1. Introduction

COVID-19, a coronavirus-transmitted infectious disease, was first identified in Wuhan, China, and declared a pandemic by the World Health Organization on 11 March 2020 [1,2,3]. The pandemic spread of COVID-19 is an undefined medical challenge and unprecedented measures have been taken worldwide. Moreover, the COVID-19 pandemic placed an enormous burden on the global healthcare system and had a substantial impact on patients with chronic diseases, making their follow-up appointments and surveillance more difficult [4,5,6,7,8,9]. The COVID-19 pandemic had a profound effect on vulnerable populations and, in Romania, it has continued to spread more and more until the present day [1].

The COVID-19 pandemic has had a deeply negative impact on every aspect of patients’ daily lives. Moreover, many of COVID-19’s risk factors are also frequently diagnosed comorbidities of obstructive sleep apnea (OSA), which has become a highly prevalent sleep-related breathing disorder. These comorbidities are associated with high mortality in patients with COVID-19 and include arterial hypertension, coronary heart disease, hyperlipidemia, type II diabetes mellitus, and obesity, being responsible for more severe disease and worse prognosis. In addition, for those with low incomes, in terms of COVID-19 disease, lower socioeconomic status has been associated with severe illness and increased mortality [1,10,11,12].

OSA is defined as a sleep disorder that involves the cessation or a significant decrease in expired airflow in the presence of breathing effort. It is the most common type of sleep-disordered breathing and is characterized by recurrent episodes of upper airway collapse during sleep. These episodes are associated with recurrent oxyhemoglobin desaturations and arousing from sleep [13,14]. OSA is diagnosed if the apnea-hypopnea index (AHI) is greater than or equal to 15 times per hour, or between 5 and 14.9 events per hour, with documented symptoms of unintentional sleep episodes during wakefulness, daytime sleepiness, insomnia, mood disorders, loud snoring, breathing interruptions during the patient’s sleep, or documented hypertension, ischemic heart disease, or a history of strokes [15,16,17].

These sequences of obstructive events, such as apnea or hypopnea, are responsible for a high oxidative stress level and for sympathetic activation, which is involved in the determination of comorbidities associated with OSA. Moreover, it is a well-known fact that the prevalence of these comorbidities is directly correlated with elevated levels of mental distress. It could be argued that the additional mechanisms observed in OSA patients, such as inflammation, oxidative stress, or immune system function issues, could be involved in this link [18,19].

In addition, OSA symptomatology affects day-to-day life, as OSA increases the risk of traffic accidents, being associated with excessive daytime sleepiness (EDS) in approximately 50% of patients [20,21]. Even if the reports differ, in most cases, the driving risk in OSA is more closely related to the degree of daytime sleepiness than to the objectively measured severity of sleep-disordered breathing [22,23,24]. Effective OSA treatment, usually with continuous positive airway pressure (CPAP), rapidly reduces both the apnea-hypopnea index and excessive daytime sleepiness in most of the affected patients, leading to a reduced number of road crashes [25,26,27,28]. These facts have led to a revision of Annex III of the European Union (EU) Directive on driving licenses, which is subject to mandatory implementation by all member states from 31 December 2015 and has become important in Romania as well [29,30,31].

In addition, in the current context, which is the rapid spread of infection, the number of deaths caused by COVID-19, the imposition of home confinement for indefinite periods of time, and the growing financial losses incurred can convey an increased risk of psychiatric conditions among all layers of society, which will also adversely affect the risk of car crashes. Moreover, several studies on many related subjects have been published, showing a high prevalence of anxiety, stress, depression, and post-traumatic stress disorder (PTSD) in healthcare workers, a higher risk of distress, anxiety, depression, and sleep disturbance in nurses, and a sevenfold increase in depression rates in the general population [32,33].

Depression and anxiety have a clinically relevant association with OSA, and, as Lee et al. observed in a female cohort, excessive daytime sleepiness and lower education levels were related to anxiety [34]. OSA patients with depression could experience higher levels of fatigue and lower quality of life than OSA patients without mood disorders. A recent meta-analysis showed relatively high rates of symptoms of anxiety (ranging from 6.3% to 50.9%), depression (14.6–48.3%), post-traumatic stress disorder (7–53%), psychological distress (34–38%), and stress (8.1–81.9%) in the general population across the globe during the COVID-19 pandemic. Anxiety and depression are known to have reciprocal relationships with insomnia [10,35] and it has been shown that the prevalence of all forms of psychological distress in the general population has been higher during the pandemic [36,37]. In these conditions, depression and anxiety could also make CPAP therapy difficult in OSA patients. This suggests that undiagnosed or untreated OSA patients, those with lower continuous positive airway pressure compliance, and those with mood disturbances may be at higher risk of developing severe forms of COVID-19 than the general population [38,39].

Anxiety is a normal reaction to a stressful situation and the response to supportive interventions and coping strategies is generally positive. For example, increased anxiety levels during the pandemic are associated with fuller compliance with governmental measures and hygienic practices. Moreover, during pandemic times, anxiety regarding health matters can rapidly become excessive. For these subjects, this stressful situation can determine anxious behavior (repeated medical consultations, avoiding health care even if needed, etc.); in the general population, it can lead to mistrust of public authorities’ safety measures or non-adherence to infection control strategies and the stigmatization of certain groups [32,40].

In the context of the risk of car crashes and depression or anxiety, during the COVID-19 pandemic it was observed that, of all road accidents, there was an increased number of speed-related crashes and fatalities. Vingilis et al. identified several potential factors that affected road safety during the pandemic, with personal factors such as a propensity for risky behavior, along with situational factors such as fuel price increases and the improper application of laws [41]. We must also consider that, during the pandemic, high levels of stress, depression, and anxiety were reported in some population groups, while increased alcohol sales and use were observed [42,43,44]. With these well-known factors affecting road crashes, the multifactorial impact of the pandemic on road safety requires an interdisciplinary approach [45].

To determine the anxiety or depression status of OSA patients, the stress perception questionnaire (PSQ), developed by Levenstein [46], is a useful multiple-choice questionnaire (Appendix A).

There were 3 main objectives of the current study:(1)To evaluate how patients managed their sleep apnea during the COVID-19 pandemic, considering their symptomatology and comorbidities and the higher risk of severe disease with a fatal outcome;(2)To establish if CPAP adherence has increased/decreased after the onset of the pandemic;(3)To compare the stress level in OSA patients with their pre-pandemic levels and to observe if its modification is related to their individual characteristics (age, gender, and BMI) or to comorbidities and apnea severity.

The aim of the study was to assess the level of stress and anxiety in OSA patients during the COVID-19 pandemic and to compare them to the pre-pandemic period, to quantify the subjects’ characteristics (the severity of OSA and their comorbidities), their impact on individual perceptions regarding COVID-19 infection, and to quantify if the stress or other factors (age, gender, environment, individual perception about COVID-19, and the level of knowledge) influence their CPAP compliance. This paper studies the impact of the COVID-19 pandemic on patients from Transylvania, who were diagnosed with OSA in the Sleep Laboratory of the “Iuliu Hatieganu” University of Medicine and Pharmacy, regarding their stress level and, as a consequence, their CPAP usage. Information obtained from this survey indicated if OSA subjects who were treated at home with a CPAP device were careful enough to ensure that their disease was being effectively controlled during the pandemic. The study also took into consideration the degree of awareness of OSA patients that they are more susceptible to COVID-19 and are more likely to develop more severe complications of the disease.

The present article is organized with an introduction, materials and methods, results, a discussion, and our conclusions. Each chapter has subchapters pointing out the parameters for analysis and the main findings of the study.

## 2. Materials and Methods

### 2.1. Study Design and Setting

Between 16 March and 14 May 2020, we conducted a retrospective observational study in the Sleep Laboratory of the “Iuliu Hatieganu” University of Medicine and Pharmacy. The study included patients diagnosed with OSA from September 2019 to November 2019 who underwent CPAP therapy at home.

At their diagnosis, all participants underwent a cardiorespiratory sleep study using a VitalNight Pro polygraph device, which incorporates continuous recordings from a nasal cannula, heart rate, oxygen saturation, tracheal sounds (microphone), thoracic and abdominal movement, and body position. The sleep study results were analyzed and approved by trained personnel. CPAP titration was made using a CPAP device (Philips Respironics Dream Station Auto CPAP) after a validated protocol [47]. The initiation of therapy was indicated according to Medicare guidelines, as follows: all patients with an AHI greater than 15 were considered eligible for CPAP, regardless of symptomatology; for patients with an AHI of 5–14.9/h, CPAP was indicated only if the patient had one of the following symptoms: excessive daytime sleepiness, impaired neurocognitive function, mood disorders, insomnia, cardiovascular disease (e.g., hypertension or ischemic heart dis-ease), or a history of stroke [6]. After the initiation of CPAP therapy, the patients underwent a medical visit one month after the titration, which included reading the CPAP therapy device cards to evaluate the residual AHI and therapy compliance.

During the national state of emergency (SOE), a telephone questionnaire survey of 46 OSA patients was conducted, and telemedicine was used for the first time in the management of OSA subjects in our center. Patients from the Transylvania region who had been diagnosed in our department were contacted by us, either for a follow-up visit or to examine the progress of their OSA management. Patients offered their written consent and the documents were sent to them via e-mail. After they agreed to participate in the form of a video or telephone call, the patients returned the signed documents and they were enrolled in the study.

Moreover, for those OSA subjects included in the survey, information was offered to ensure that the CPAP device was being used effectively, and that adequate supplies were available, with appropriate masks, tubing, and CPAP machine sanitization, to encourage regular adherence to nightly CPAP use, and that patients were following the Centers for Disease Control and Prevention guidelines.

The data recorded by the CPAP devices were downloaded by our sleep technicians, using VitalNight EasyScore software version 5.22afrom the database wherein the information from patients’ CPAP devices is stored via telehealth services and was interpreted according to the guidelines [13] issued by the sleep-medicine doctors involved.

### 2.2. Participants

During the study period, a total of 108 patients who were diagnosed in our clinic were contacted.

The inclusion criteria for patients were an age ≥ 18 years, a diagnosis of OSA established between September 2019 and November 2019, with CPAP therapy at home for at least three months, and, at the moment of the survey, to be undergoing CPAP therapy. We considered that patients who do not meet the inclusion criteria were excluded from the study. The exclusion criteria were that the patients should be aged ≤18 years, with a diagnosis of OSA established before September 2019 or after November 2019, and those patients diagnosed with OSA who were not using CPAP therapy at home.

After applying the inclusion criteria, a cohort of 46 patients was obtained. From the total number of patients contacted, 24.07% (26 subjects) had abandoned CPAP therapy of their own free will (from their diagnosis of OSA until the time of the survey), 25% (27 subjects) could not be reached (9 of them had moved from the city, 10 changed their phone numbers or did not answer, and 8 had died) and 8.3% (9 patients) did not offer their informed consent (Figure 1).

All subjects gave their informed consent for inclusion before participating in the study. The study was carried out in accordance with the Declaration of Helsinki [48] and the protocol was approved by the Ethics Committee of the “Iuliu Hatieganu” University of Medicine and Pharmacy Cluj-Napoca, with the reference number 270/02 February 2020.

### 2.3. Variables

The collected data (Table 1) comprised personal information (sex, age, environment, and smoker status), anthropometric measures (weight, height, BMI, neck circumference, and abdominal circumference), associated diseases (hypertension, chronic ischemic cardiopathy, myocardial infarction, dyslipidemia, cardiac failure, diabetes mellitus, asthma, and COPD), sleep parameters collected at diagnosis according to the database (AHI, ODI, SaO_2_ minimum, average SaO_2_, and nocturnal and diurnal symptomatology), Epworth sleepiness scale rating (at diagnosis, one month after the diagnosis, and during the pandemic) (Appendix A), CPAP parameters one month after the diagnosis, according to the database, and during the pandemic (average time of use, compliance above 4 h, residual AHI), and PSQ score (at baseline and during the pandemic).

### 2.4. Data Sources

As a classical instrument to evaluate stress levels, we used the perceived stress questionnaire (PSQ), an assessment used as a routine measurement tool in our laboratory and during the pandemic period, along with an original questionnaire designed for OSA patients that were on CPAP treatment. The perceived stress questionnaire is used as an instrument to assess the stressful life events and circumstances that tend to trigger disease symptoms [46]. With stress significantly affecting the quality and consistency of the sleep cycle, the PSQ is a potentially valuable tool to evaluate the underlying cause of sleep disturbances. To complete the PSQ, respondents receive one of two sets of scoring instructions: the general questionnaire queries stressful feelings and experiences over the course of the previous year or two, while the more recent questionnaire concerns stress during the previous month. Respondents indicate on a scale of 1 (“almost never”) to 4 (“usually”) how frequently they experience certain stress-related feelings. Higher scores indicate a greater level of stress. The interviewer must remind the subjects that they must circle the number that describes how often the sentences applied to their situation in the last month. A total score is found by tallying each item (questions 1, 7, 10, 13, 17, 21, 25, and 29 are positive and are scored according to the directions accompanying the scale). A score between 30 and 59 is classified as reduced stress, between 60 and 89 is classified as moderate stress, and between 90 and 120 is classified as high stress (Appendix A).

The original questionnaire was conceived considering our experience with COVID-19 patients and included 7 questions regarding the individual’s COVID-19 status, the modifications of the daily schedule (working from home), the subjective considerations about feeling depressive or anxious, the changes in sleep schedule and sleep quality and the factors that can influence it, and the individual’s perception of the risk of contracting COVID-19 (Appendix A).

### 2.5. Statistical Analysis

Statistical analysis was performed using MedCalc^®^ Statistical Software, version 20.014 (MedCalc Software Ltd., Ostend, Belgium; https://www.medcalc.org; 2021, accessed on 16 October 2022). Quantitative data were examined for normality of distribution, using the Shapiro–Wilk test, and were expressed as mean ± standard deviation or median and 25th–75th percentiles. Qualitative data were expressed as frequency and percentage. Regarding the changes in CPAP compliance (nights with CPAP usage of over 4 h) during the COVID-19 pandemic, compared to one month after diagnosis, for our sample size, we calculated the power of the study to be at 99% (α = 0.01) for a level of significance of 1% (β = 0.01). Comparisons between groups regarding qualitative variables were performed using the chi-square test. Comparisons between groups regarding quantitative variables were performed using the Mann–Whitney test. Correlations between variables were verified using Spearman’s rho, a non-parametric test used to measure the strength of association. The Wilcoxon test, a non-parametric test, was used to evaluate the change of a variable between two repeated measurements. A *p*-value of lower than 0.05 was considered statistically significant.

## 3. Results

### 3.1. General Characteristics at Baseline of the Included Patients

The information was collected from patients diagnosed with OSA from September 2019 to November 2019. We chose this period because, after three months of home CPAP therapy, the doctor can determine the individual pattern of CPAP for the patient, so the adherence parameters are more stable [49].

The mean age of the studied population was 56.7 years, ranging between 37 years and 76 years; 80.4% of the subjects were males and 19.6% were females. In addition, considering the smoker status, 28.3% of the subjects were non-smokers and 71.7% were active smokers.

One of the most important comorbidities was obesity, so the mean body mass index (BMI) in our study group was 36.33 (33.42–41.21) kg/m^2^, with most of the patients being obese. In addition, the anthropometric measurements recorded at diagnosis indicate a median neck circumference of 47.4 ± 4.3 cm and a median abdominal circumference of 124.5 (115.75–141.00) cm, suggesting central disposition of the adipose tissue, which predisposes patients to the development of OSA.

Most of the included patients also presented associated pathologies correlated with OSA; 78.3% had hypertension, 47.8% had dyslipidemia, 10.9% of the patients had suffered cardiac failure, and 4.3% of the patients had a myocardial infarction prior to the evaluation, leading to high morbidity and mortality levels in the context of SARS-CoV-2 infection. Of the patients, 23.9% were diabetic; 17.4% were treated with oral antidiabetic medication, 4.3% were insulin-dependent, and 2.2% needed insulin, a disease with a high impact on SARS-CoV-2 patients’ evolution. Moreover, 10 patients presented chronic respiratory diseases: 4.3% had asthma, and 17.4% suffered from COPD (Table 2).

We included the PSQ questionnaire as a routine evaluation; the mean score at diagnosis was 54 (41.75–66.25), with 17 (36.9%) of the subjects having a minimum stress level and 29 (63.1%) having a moderate stress level (Table 2).

### 3.2. The Sleep Parameters at the Baseline

As seen in Table 3, we quantified the nocturnal respiratory parameters at the time of diagnosis. The mean AHI was 63.75 (39.9–81.2) events/hour of sleep and the desaturation index was 59.6 (38.8–79.1) events/hour of sleep, indicating that most of the subjects have severe OSA. In addition, the minimum oxygen saturation was 65% (60–74.3%) with a mean oxygen saturation of 87.5% (81.8–90.3%), showing that the patients have important nocturnal desaturations. We also quantified the presence of nocturnal and diurnal symptoms at the time of diagnosis; 95.7% reported snoring, 65.2% had apnea episodes during the night that had been observed by their family, 15.2% had nightmares, 78.3% had nocturia, 80.4% reported daytime sleepiness, 45.7% reported a morning headache, 67.4% reported morning fatigue, 43.5% reported that it influenced their work capacity, those being the most frequent symptoms in people with sleep apnea. The Epworth sleepiness scale showed a median value of 16 (12–19.25) and, in 82.6% of subjects, the total score showed values above 10 points; therefore, in 38 cases, we identified excessive daytime sleepiness (Table 3).

### 3.3. CPAP Compliance Parameters at One Month after Diagnosis

One month after diagnosis, the patients diagnosed with OSA performed a routine evaluation. By reading their CPAP cards data, we analyzed their CPAP compliance and concluded that the average duration of use was 354 (288.7–389.2) min/night, their compliance above 4 h was 69.5 (54.7–76.0)% and their residual AHI was 5.6 (3.35–9.3) events/hour of sleep (Table 4). As seen in Figure 2, 28.26% had a residual AHI above 5 events/hour of sleep, 52.18% had between 5–14.9 events/hour of sleep, and 19.56% had a residual AHI above 15 events/hour of sleep, one month after their OSA diagnosis. As seen in Table 4, we followed the patients’ CPAP compliance and compared it one month after their diagnosis to one month during the pandemic. The average use per night of CPAP increased from a median of 354.5 min one month after their diagnosis to 399.5 min during the state of emergency, with statistical significance (*p* = 0.00). As a result, compliance above 4 h showed an improvement from a median value of 69.5% one month after diagnosis to 79% during the state of emergency (*p* = 0.000). This finding leads to a reduced number of events per night, from a median of 5.6 one month after diagnosis to 2.4 during the state of emergency (*p* = 0.000). CPAP compliance during the state of emergency increased in the case of patients diagnosed with a COVID-19 infection. Those subjects had a median compliance of 73% (58.75–79.00%) one month after their OSA diagnosis and of 82% (72–84%) during the state of emergency. In addition, CPAP compliance during the state of emergency increased in the case of patients with COVID-19. In addition, the patients increased their CPAP usage from 363 (311–402.75) min/night to 429 (398.75–464.5) min/night (*p* = 0.012).

As mentioned before, we evaluated the patients using a questionnaire comprising 7 questions. Regarding question no. 1 of our evaluation, 43.47% of the subjects were diagnosed with COVID-19 at the beginning of the pandemic (70% men, 30% women) and of those, 70% presented important symptomatology, more frequently showing fever, cough, shortness of breath, myalgia, and expectoration. Of all subjects infected with SARS-CoV-2, 80% had arterial hypertension, 75% had diabetes mellitus, and 25% had dyslipidemia, these being the most prevalent comorbidities. This fact is very important since those comorbidities are the most prevalent in OSA subjects.

In addition, considering question no. 2 of the questionnaire, 56.52% of the participants had at least one member of the family diagnosed with COVID-19; 80.4% of the subjects of the study population lived isolated from their family during the state of emergency, which had a profound impact on their mental state.

Regarding question no. 3 about each subject’s financial situation, during the state of emergency, 47.8% of patients worked from home and 17.4% became unemployed, with the rest of the study group already being unemployed at that time. Even if, of the entire population, 45.65% of the subjects reported that they had experienced symptoms of depression, especially a lack of motivation and concentration, and 36.95% experienced anxiety, in the case of those patients that lost their jobs, subjective depression symptoms were reported in 100% of cases (*p* = 0.000, compared with those who worked from home) and 75% had worries about financial loss and experienced nightmares, but not anxiety. In addition, we identified a strong correlation between unemployment and the modification of sleep patterns. Therefore, all the patients who lost their jobs presented a changed sleep schedule: 75% of the patients slept less and 25% slept more.

Regarding question no. 4 of our evaluation, 82.6% reported a sleep schedule that had been modified in the last 3 months; from those, 50% had a COVID-19 infection. Other complaints concerning sleep quality were as follows: 13.1% of cases experienced difficulties in maintaining sleep, 13.1% had subjective excessive daytime sleepiness, waking up early was reported in 8.7% of the cases, and 4.3% stated that they had unrefreshing sleep. The usage of different substances (question 4b) to improve the subjective quality of sleep was analyzed. While melatonin, cannabis, or other substances were used in reduced proportions, we observed that 52.2% of the patients included in the study had used alcohol to obtain a better quality of sleep.

In terms of question no. 5 (both a and b) regarding the high prevalence of obesity, we analyzed weight fluctuations during the state of emergency, compared to basic values, and the correlation with different parameters. Therefore, 84.8% of the patients reported that their weight was modified in the last three months; of these, 64.1% gained weight, with an average weight gain of 6.24 kg, while 35.9% lost weight, with an average weight loss of 6.78 kg. Conversely, 11% reported that their weight was constant, while 4.2% did not know how to answer the question. The correlation between weight and compliance with CPAP was approached. As a result, 52.3% of the patients that gained weight used their CPAP more, mostly because of the accentuation of OSA symptoms, compared with only 28.57% of those who had lost weight. Regarding weight gain and sleep schedule modifications, 84.61% of the patients who gained weight had a modified sleep pattern and claimed unrefreshing sleep, subjective daytime sleepiness, and morning fatigue.

On collating the answers to question no. 6, we concluded that 73.9% of the subjects felt fear regarding COVID-19 infection, while 84.8% considered that they had an increased risk of COVID-19 infection. From the total number of subjects, only 65.2% of the included people had information about the pandemic, obtained in equal proportions from the Internet, television, and from their attending doctor.

Regarding question no. 7 on the state of emergency, the emotional aspect of OSA patients was influenced in 34.7% of cases by worries about themselves, 39.13% were affected by loneliness, 19.56% experienced nightmares, 17.39% were influenced by worries about financial loss, and 10.86% were influenced by worries about family/friends, with most of the patients experiencing more than one emotional problem.

We also analyzed the Epworth sleepiness score during the state of emergency and we observed that only 39.1% of the patients currently had a score above 10, compared to 71.3% at the time of diagnosis. Moreover, the median value of the Epworth score at diagnosis was 16 versus 5 during the state of emergency (*p* = 0.00) and the Epworth score at diagnosis correlated with compliance above 4 h during the state of emergency (*p* = 0.02).

In addition, as stated before, the perceived stress questionnaire (Appendix A) was applied two times, at the time of diagnosis and at another evaluation during the emerging cases of COVID-19, between 16 March and 14 May 2020; this was included in the patient’s medical records. The median value for the stress perception scale before the state of emergency was 54 and increased during the state of emergency to 93, a difference with clinical significance (*p* = 0.000). During the state of emergency, 17.39% of the subjects were in the lower stress category, 30.43% in the moderate stress category, and more than half of the subjects, 52.17%, were in the intensive stress group. Regarding the weight modification and stress perception scale categories, both of these being factors with implications for CPAP compliance, a correlation was made; 75% of the patients included in the lower stress category lost weight. For the medium level of stress, 2 (14.3%) lost weight, 3 (21.4%) maintained a constant weight, and 9 (64.3%) gained weight. In the category of a high level of stress, 6 (25%) of the patients were represented by those who lost weight, 3 (12.5%) maintained the same weight, and 15 (62.5%) gained weight. Moreover, the anxiety and depression modification percentages may be influenced by the fact that we included only 9 females in our study; this is significant because they are underdiagnosed with OSA due to the lack of classic symptoms. Studies that include a greater number of patients need to be developed, in order to clarify if the emotional changes are more important in the female population of OSA patients.

As stated in Table 5, we observed that compliance above 4 h before the state of emergency was negatively correlated with PSQ at the time of diagnosis (*p* = 0.001), AHI (*p* = 0.006), and desaturation index (*p* = 0.000), and was positively correlated with the minimum saturation of oxygen (*p* = 0.40). Events per hour of sleep one month after diagnosis were correlated with AHI (0.031) and desaturation index (*p* = 0.012), and were indirectly correlated with the minimum saturation of oxygen (*p* = 0.030).

Average use of AutoCPAP therapy per night during the state of emergency correlated with AHI (*p* = 0.001), the desaturation index (*p* = 0.020), and PSQ evaluated during the state of emergency (*p* = 0.002).

Taking into consideration that there might be other factors that influence the usage of CPAP, we evaluated the correlation between compliance before and during the state of emergency, along with the grade of apnea or level of stress. Therefore, a statistically significant correlation was identified between compliance during the state of emergency and AHI at the time of diagnosis (*p* = 0.049). Moreover, the correlation between compliance during the state of emergency and the PSQ score during the state of emergency is also statistically significant (*p* = 0.014).

## 4. Discussion

The COVID-19 pandemic brought a new challenge to healthcare systems worldwide. The concern for patients with comorbidities was founded on high morbidity and mortality levels in this category. Although there are currently few studies regarding the prevalence of COVID infection in patients who had previously been diagnosed with OSA, the data show that the presence of this pathology increases the risk of a negative outcome in COVID-19 patients [4,50,51,52].

Furthermore, infection with the new virus leads to the development of an aggressive type of pneumonia with a high risk of mortality in elderly patients, particularly in those with comorbidities, such as diabetes, obesity, and hypertension [4,5,12,53]. The association between OSA and COVID-19 is currently undergoing research. To evaluate the impact of the COVID-19 pandemic on an OSA patient’s status, we conducted a study that included 46 subjects diagnosed with OSA who received CPAP treatment at home for at least three months before the beginning of the pandemic, because the usage pattern and the adherence parameters were more stable at this period of time.

The treatment of OSA patients and the use of CPAP devices remain major challenges for healthcare professionals. Good adherence and the proper use of CPAP devices are the main elements in the successful management of disease in these patients [10,54].

OSA is a highly prevalent disease that occurs in 24% of young to middle-aged men and 70% of older men, in 9% of young women, and in 56% of older women [55]. The results are comparable with those of our study, in which the disease was found in 86% of cases in men. It has been postulated that the higher clinical ratio may be a result of the fact that women do not show the “classic” symptomatology and, thus, may be underdiagnosed.

Obesity has long been known to be associated with OSA; for both genders, the body mass index (BMI) correlates positively with the severity of the disease [56,57]. This hypothesis is supported by our study, the mean BMI of our cohort being 36.33 kg/m^2^, with a mean neck circumference of 47.4 ± 4.3 cm and a median abdominal circumference of 124.5 cm, indicating a central disposition of the adipose tissue.

In addition, as Bonsignore et al. showed, OSA patients show a high prevalence of cardiovascular diseases (systemic hypertension, coronary artery disease, arrhythmias, and ischemic stroke), respiratory diseases (COPD and asthma), and metabolic disorders (diabetes mellitus, dyslipidemia, and gout) [58]. In our study, 78.3% of patients had systemic arterial hypertension, 47.8% had dyslipidemia, 10.9% suffered cardiac failure, 4.3% suffered a myocardial infarction, 23.9% were diabetic, and 21.7% had respiratory diseases (4.3% had asthma, and 17.4% had COPD). OSA and COVID-19 disease share common comorbidities; studies showed that many patients who had developed a severe form of COVID-19 infection also had diabetes, systemic arterial hypertension, and other respiratory diseases [59]. The profile of patients at greater risk of developing complications or even of death associated with COVID-19 infection [4,53,54,60] is similar to our population of OSA patients, many of them being diagnosed with obesity (75%), hypertension (80%), dyslipidemia (25%) and other associated respiratory pathologies.

Moreover, we included the PSQ questionnaire as a routine evaluation; the mean score at the time of diagnosis was 54 (41.75–66.25), with 17 (36.9%) subjects having a minimum stress level and 29 (63.1%) having a moderate stress level. As shown in the study by Celik et al., OSA patients are predisposed to a significant level of stress, which is reduced while using CPAP [61].

At the beginning of the COVID-19 pandemic, we elaborated an original questionnaire (Appendix A) which showed that 43.47% of the OSA subject sample was infected with COVID-19 and had important symptomatology. In addition, 56.52% of the participants had had at least one family member diagnosed with COVID-19, and most of the patients lived in isolation. In the current situation of the global pandemic, studies show that OSA patients are generally aware of their supplementary risk level when compared to the general population [35,62]. The results of our research show similar results, taking into consideration that most of the patients lived in isolation from their families, in order to minimize the spread of the virus through CPAP-generated aerosols. Furthermore, most of these subjects had more than one comorbidity, increasing their risk of developing a severe form of COVID-19 infection, as stipulated in the literature [59].

In our study, most of the patients had severe OSA, with a mean AHI of 63.75 (39.9–81.2) events/hour of sleep, a highly important aspect considering that severe and moderate OSA becomes a risk factor when a patient is diagnosed with COVID-19, as the literature reveals [63]. The patients’ associated symptoms, such as snoring, apnea episodes reported by the family, daytime sleepiness, morning fatigue, and morning headache influenced work capacity, nightmares, and nocturia. In addition, as the literature shows, the main symptoms seen in OSA patients include excessive daytime sleepiness, non-refreshing sleep, fatigue, morning headache, memory loss, nocturia, and irritability. Untreated OSA is also associated with a lack of concentration leading to a loss of productivity in the workplace and road crashes, as previously shown in many studies, that resulted in fatality [21,24,64]. Even if we did not take into consideration the risk of road crashes, it is important to mention that the coexistence of untreated OSA and depression or anxiety during the COVID-19 pandemic has become an important risk factor for these events.

Data from question 3 showed that 17.4% of the subjects lost their jobs and 47.8% worked from home, the rest of the study population already being unemployed at that time. The patients that lost their jobs or worked from home reported symptoms of depression, especially a lack of motivation and concentration, along with anxiety, factors that have a substantial influence on the subjects’ mental health. These data are similar to those from the literature, which stipulated that financial and social concerns resulting from the pandemic may increase rates of insomnia and anxiety during a lockdown [65]. In the case of the subjects who lost their jobs, a change in sleep pattern was observed; 75% slept less and 25% slept more, a finding that is supported by the specialized literature, which showed that during the lockdown, many subjects had irregular sleep schedules, less exposure to daylight, less physical activity, depression, anxiety, and more screen time, with many reporting lower sleep quality despite, perhaps, longer sleep duration [65].

We analyzed the weight fluctuation patterns during the state of emergency and observed that 64.1% of the patients gained weight (6.29 kg median) and 35.9% of the patients lost weight (a median of 6.78 kg). Additionally, 84.61% of the patients had a modified sleep schedule and sleep disturbances and 52.3% of them used CPAP more. It is a well-known fact that patients with mild OSA who gain 10% of their baseline weight are at a sixfold-increased risk of progression of OSA, while an equivalent weight loss can result in a more than 20% improvement in OSA severity. In addition, obesity may negatively influence the control of OSA because of fat deposition at specific sites in the body, which is the reason why OSA symptoms are more frequent and important, meaning that the patients feel more confident in using their CPAP devices [66,67].

Furthermore, our patients were adherent to CPAP treatment during the time of the COVID-19 pandemic because they were concerned about complications due to COVID-19. In addition, a significant percentage, 63%, of the patients increased their usage of CPAP during the pandemic. The presence of a high level of anxiety and depression, as well as increased worry concerning contracting the virus, may be a possible explanation for their good compliance with the treatment of OSA.

Data from the literature show that OSA patients have an increased level of stress regarding COVID-19 infection and the possible outcomes of the disease, an aspect that influences their mental health and their behavior [68,69]. The studies showed that high levels of stress during the state of emergency correlated with increased CPAP adherence and an improved outcome and management of OSA, with fewer sleep events [68]. Moreover, a study from France, which included 7485 OSA patients and CPAP users and lasted for a year and four months, has also demonstrated that CPAP adherence increased remarkably during the lockdown period (3.9%, *p* < 0.001). Moreover, when comparing the data with the same period in 2019, from 15 March to 15 April, they stated that very low adherence to CPAP therapy decreased from 18.5% in 2019 to 4.48% in 2020 (*p* < 0.001) [68]. Our data showed that the level of stress increased significantly during the state of emergency with a median PSQ value of 54, versus 9.3 before the pandemic (*p* = 0.000). Moreover, the average use per night (354.5 min vs. 399.5 min, *p* = 0.000) and compliance above 4 h/night (69.5% versus 79%, *p* = 0.000) increased significantly. The number of events per night (5.6 versus 2.4, *p* = 0.000) decreased significantly. Therefore, better adherence to the treatment was obtained during the state of emergency; these data offer useful information suggesting that OSA patients were triggered to obtain greater control over their OSA during the COVID-19 pandemic [68,70].

Of our studied population, 82.6% reported sleep disturbances, 45.7% reported signs of depression, and 37% reported anxiety. In total, 71.7% were worried about contracting the new virus. Multiple data from the literature reported sleep disturbances during the COVID-19 outbreak in patients with OSA [69,71,72]. Still, the improvement of the median Epworth score before and during the state of emergency (16 versus 5), despite the sleep disturbances reported by the patients, is an additional factor in favor of increased adherence to the CPAP regimen (the score correlated with compliance above 4 h, *p* = 0.02). The correlation between a lack of sleep and depression and anxiety was proven [69]. Mood disturbances and modified sleep schedules have become issues that worsen during a public health crisis, such as in the COVID-19 pandemic [73]. As Grubac et al. showed in two studies on rats, the duration of sleep fragmentation is a significant determinant of anxiety-linked behavior and, thus, may also have influenced the anxiety level in our cohort [74,75].

Multiple sleep societies have formulated the advice that OSA patients who use CPAP devices should self-isolate if they have COVID-19-compatible symptoms. Our results show that 72% of those patients with members of the family who tested positive slept in different rooms. Still, the recommendation was not feasible during the stressful pandemic period [76].

One study from New York, which included 122 patients using CPAP therapy, reported that most of the subjects included did not believe that they had an increased risk of contracting COVID-19 or could suffer a more complicated outcome. Still, 88% of the patients continued to use their CPAP device [77]. From our studied population, 84.8% of the subjects thought that they had an increased risk of COVID-19 infection. Moreover, 100% of the patients included in our study continued to use their CPAP devices. These differences may be explained by the different spreads of the disease and a later maximum incidence in the USA, compared to Europe. As a result, in the abovementioned study, 9% of the patients tested positive for coronavirus infection [78], while 43.5% of the patients were COVID-19 positive.

Treatment and the efficient management of OSA patients during the COVID-19 outbreak was a major challenge for healthcare specialists, mainly because of the closing down of most of the sleep centers. The opportunity to communicate with patients via telemedicine offered better control over the situation by encouraging many OSA patients to control their symptoms, adjusting the settings of the ventilator, and offering therapeutic indications [79,80].

The management of OSA subjects during the COVID-19 pandemic has been a challenge, and discordant statements have been made. While some authors recommend that patients with OSA should give up the use of CPAP at home during the COVID-19 pandemic [81], others have recommended that CPAP therapy be continued, especially for those with other comorbidities [82,83]. Still, most of the patients included in the present study described feeling greater health security when using CPAP therapy, reducing their level of anxiety and depression.

The lack of use of CPAP devices at home would affect their quality of life and the return of their symptoms, with an increased risk of cerebrovascular and cardiovascular diseases for OSA patients [83,84]. In addition, untreated OSA can increase the risk of car accidents, along with other factors, such as unrefreshing sleep, sleep schedule alterations, sedative medications, alcohol intake, shift work, inadequate sleep time, and poor sleep hygiene [24,25]. To obtain satisfactory results with OSA subjects during the COVID-19 pandemic, the healthcare specialist from the sleep laboratories was advised to make changes to their usual routine and keep in close contact with their patients through adapted methods. In this context, the role of the dentist becomes very important, especially because patients with certain risk factors (anomalies of the jaws (micrognathism and retrognathism) or of the soft tissues (macroglossia)) that can lead to a reduction in oropharyngeal space are followed up more intensively and their OSA can be detected earlier in order to implement therapy measures. This screening is of great importance considering the risk of OSA complications; thus, a multidisciplinary approach is needed [85].

The main limitation of our analysis was the epidemiological situation, which led to decreased access and then the closure of the Sleep Laboratory. Moreover, the small number of patients included, especially females, may bias the results by influencing the emotional change scores. The study group size was restricted by the method of communication, refusals to participate in the study, or the abandonment of the use of CPAP by patients of their own free will.

In the case of patients who need a close follow-up and chronic monitoring for their sleep disease, telemedicine can play a vital role in providing individualized treatment, which can assure good adherence to CPAP treatment. Interestingly, the Sleep Societies have implemented a unique platform by which they can offer sleep telemedicine resources, to introduce this into current clinical practice, and to guide sleep physicians in the proper facilitation of this useful and new tool across sleep laboratories [78,86].

Therefore, we conclude that the presence of the pandemic has led to a greater level of anxiety in patients with OSAS, largely due to the increased risk of developing serious complications if SARS-CoV-2 infection is contracted. Given that during the pandemic, patients experienced an unhealthy lifestyle that led to weight gain, sleep disorders, and an impaired social life, thus exacerbating the symptoms of obstructive sleep apnea syndrome, they became more aware of the harmful consequences of the disease and used CPAP therapy more.

## 5. Conclusions

Taking into consideration that during the pandemic, OSA patients’ disease management has become a challenge, it is highly recommended that sleep centers continue their activity and develop new approaches to diagnosing and treating sleep apnea patients. Close communication between medical professionals and people suffering from sleep apnea must be maintained via new methods such as telehealth, preventing the interruption of patient care and avoiding unnecessary travel. Moreover, those centers that were unable to operate during the pandemic should try to establish procedures to recommence sleep apnea management after the COVID-19 peak as quickly as feasible, depending on the local COVID-19 conditions.

## Figures and Tables

**Figure 1 ijerph-20-04313-f001:**
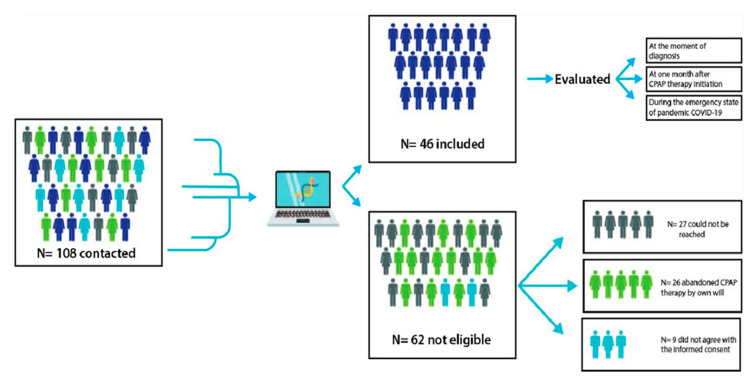
Study Design Flowchart.

**Figure 2 ijerph-20-04313-f002:**
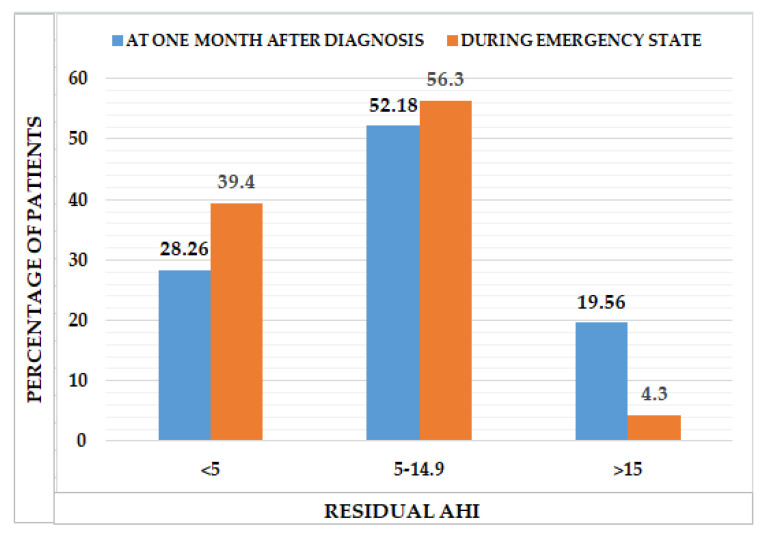
Residual AHI at one month after diagnosis and during the state of emergency.

**Table 1 ijerph-20-04313-t001:** Included variables in the data.

VARIABLES	
Category	Variables Included
Personal information	Sex
	Age
	Environment
	Smoker status
Anthropometric measures	Weight
	Height
	BMI
	Neck circumference
	Abdominal circumference
Associated diseases	Hypertension, chronic ischemic cardiopathy, myocardial infarction, dyslipidemia, cardiac failure
	Diabetes mellitus
	Asthma, COPD
Sleep parameters collected at the diagnosis	AHI, ODI, minimum SaO_2_, average SaO_2_
	Nocturnal and diurnal symptomatology
Epworth sleepiness scale - Collected At the diagnosis, one month after the diagnosis, and during the pandemic	Appendix A
CPAP Parameters - Collected one month after the diagnosis according to the database and during the pandemic	Average time of use, compliance above 4 h, residual AHI), PSQ score (at baseline and during the pandemic)
PSQ Score - at baseline and during the pandemic	Appendix A

**Table 2 ijerph-20-04313-t002:** General characteristics of the included patients at baseline.

Criteria	Response	Value (*n* = 46, % *n*)
**Place of residence**	rural	23 (50%)
	urban	23 (50%)
**Sex**	M	37 (80.4%)
	F	9 (19.6%)
**Age (years)**	min	37
	max	76
	median	56.69 ± 10.9
**Weight at diagnosis (kilograms)**	min	73.00
	max	168.00
	median	110 (95.75–125.5)
**Height** **(centimeters)**	min	158.00
	max	188.00
	median	171.6 ± 6.7
**BMI at diagnosis** **(kg/m^2^)**	min	28.00
	max	58.50
	median	36.33 (33.42–41.21)
**Neck circumference** **(centimeters)**	min	38.00
	max	56.00
	median	47.4 ± 4.3
**Abdominal circumference** **(centimeters)**	min	93
	max	158
	median	124.5 (115.75–141.00)
**Smoker status**	non-smoker	13 (28.3%)
	smoker	33 (71.7%)
**Perceived Stress Questionnaire**	min	30
	max	82
	median	54
	25th–75th percentiles	41.75–66.25
**Associated diseases**	hypertension	36 (78.3%)
	chronic ischemic cardiopathy	0(0%)
	myocardial infarction	2 (4.3%)
	dyslipidemia	22 (47.8%)
	cardiac failure	5 (10.9%)
	diabetes mellitus	11(23.9%)
	asthma	2 (4.3%)
	COPD	8 (17.4%)

**Table 3 ijerph-20-04313-t003:** The sleep parameters at the time of diagnosis (baseline).

AHI (Events/Hour of Sleep)	
**min**	23.4
**max**	132.0
**mean**	63.75
**standard deviation**	39.9–81.2
**DESATURATION INDEX (events/hour of sleep)**	
**min**	23.7
**max**	127.5
**mean**	59.6
**standard deviation**	38.8–79.1
**MINIMUM SAO_2_ (%)**	
**min**	35
**max**	87
**mean**	65
**standard deviation**	60–74.3
**AVERAGE SAO_2_ (%)**	
**min**	70
**max**	94
**mean**	87.5
**standard deviation**	81.8–90.3
**EPWORTH SCALE**	
**min**	3
**max**	23
**median**	16
**25th–75th percentiles**	12–19.25
**SYMPTOMS**	
**Snoring**	44 (95.7%)
**Witnessed apnea**	30 (65.2%)
**Nightmare**	7 (15.2%)
**Nocturia**	36 (78.3%)
**Ravished bed**	34 (73.9%)
**Chocking in sleep**	21 (45.7%)
**Daytime sleepiness**	37 (80.4%)
**Morning headaches**	21 (45.7%)
**Morning fatigue**	31 (67.4%)
**Influenced work capacity**	20 (43.5%)

**Table 4 ijerph-20-04313-t004:** Parameters of compliance one month after diagnosis and during the state of emergency.

		One Month after Diagnosis	During the State of Emergency
AVERAGE TIME OF USE (minutes)	min	164.00	210.00
	max	481.00	495.00
	mean	354.5	399.5
	25th–75th percentiles	288.7–389.2	352.5–448.0
COMPLIANCE ABOVE 4 HOURS (%)	min	42.00	51.0
	max	82.00	96.0
	mean	69.5	79.0
	percentiles 25–75	54.7–76.0	71.7–84.0
RESIDUAL AHI (Events/hour of sleep)	min	0.5	0.5
	max	20.00	12.1
	mean	5.6	2.4
	percentiles 25–75	3.35–9.3	1.6–7.0

**Table 5 ijerph-20-04313-t005:** Correlation between the evaluated parameters.

Parameter Correlation Coefficient-*p*	Compliance above 4 h One Month after Diagnosis (%)	Residual Events/ Hour of Sleep One Month after Diagnosis	Average Use/Night at One Month after Diagnosis (min)	Compliance above 4 h ES (%)	Events/ Hour of Sleep ES	Average Use/Night ES (min)
PSQ at the diagnosis	−0.490	0.233	−0.172	NA	NA	NA
	0.001	0.119	0.253	NA	NA	NA
Age	0.153	−0.216	−0.104	−0.212	−0.087	−0.267
	0.310 **	0.149	0.493 **	0.158	0.564 ***	0.073
AHI at the diagnosis	−0.396	0.318 **	−0.095	0.292	0.036	0.476 **
	0.006	0.031	0.531 ***	0.049	0.814 ***	0.001
Desaturation index at the diagnosis	−0.529	0.369 **	−0.223	0.140	0.193	0.342 **
	0.000	0.012	0.136	0.352 **	0.198	0.020
Minimum SaO_2_% at the diagnosis	0.304 **	−0.321	−0.049	0.081	−0.213 *	−0.265
	0.040 **	0.030	0.747 ***	0.593 ***	0.155	0.075
Average SaO_2_% at the diagnosis	0.264	−0.152 *	0.013	−0.207	0.125	−0.444
	0.076	0.314 **	0.931 ***	0.167	0.406 **	0.002
PSQ ES	NA	NA	NA	0.361 **	−0.067	0.436 **
	NA	NA	NA	0.014	0.658 ***	0.002

Correlation coefficient: 0.1–0.3—weak correlation *, 0.3–0.5—medium correlation **, > 0.5—strong correlation ***; negative—indirect correlation, positive—direct correlation; *p* < 0.05—statistically significant.

## Data Availability

Not applicable.

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
