# Peer review of "The Management of Obstructive Sleep Apnea Patients during the COVID-19 Pandemic as a Public Health Problem—Interactions with Sleep Efficacy and Mental Health"

_ijerph, 2023, doi:10.3390/ijerph20054313_

Round 1

Reviewer 1 Report

In this paper, the authors present an study aimed to evaluate the management of obstructive sleep apnea patients during the covid-19 pandemic, taking into account the sleep efficacy and mental health.

After a detailed review, I consider that this is a well-written and interesting paper, very related with the aims and scope of the IJERPH journal.

After accepting it for publication, I would like to point some questions.

Please, see the attached pdf file.

Author Response

Thank you very much for your review, which I find very useful in significantly improving our article. We have provided point-by-point answers to your observations below.

Q1. I did not find the supplementary materials. Please revise that you upload them.

Thank you very much for the observation. We checked again and uploaded the supplementary materials.

Q2. I think that it could be a good idea to summarize the variables of section 2.3 in a table.

As you suggested, we summarized the variables of section 2.3 in a table (Table 1, lines 224-225)

Q3. In Section 2.5 the authors point to several statistical methods. They should justify their election for using parametric or no parametric tests.

As we mentioned, we used the Shapiro Wilk test to determine the normality of the distribution for quantitative date.

Also, for quantitative variables we used Man-Whitney test and for the qualitative ones we used chi-square test, as is mentioned in the section 2.5 (lines 259-261).

In addition, we made modifications as you suggested, beginning with the line 262.

Q4. According to the instructions for authors of the IJERPH journal: ‘The abstract should be a total of about 200 words maximum. The abstract should be a single paragraph and should follow the style of structured abstracts, but without headings: 1) Background: Place the question addressed in a broad context and highlight the purpose of the study; 2) Methods: Describe briefly the main methods or treatments applied. Include any relevant preregistration numbers, and species and strains of any animals used. 3) Results: Summarize the article's main findings; and 4) Conclusion: Indicate the main conclusions or interpretations. The abstract should be an objective representation of the article: it must not contain results which are not presented and substantiated in the main text and should not exaggerate the main conclusions.’. Please revise the abstract and remove the headings. I recommend the authors to do those changes necessary to adapt it.

Indeed, we modified the abstract and made the changes necessary to adapt it, thank you for the suggestions.

Q5. I recommend the authors to add a paragraph at the end of the introduction section explaining how the rest of the paper is organized.

we added the paragraph at the end of the introduction section, and we consider now that is a useful one in order to be clearer, thank you for the recommendation (lines 20-34).

Q6. There are some lines in blank in the manuscript (for example 297 -299). Please revise the manuscript.

We revised the manuscript to delete the blank rows, thank you.

Q7. From Line 327 to 379 the results of the 7 questions questionnaire are presented. I think that it could help to summarize them in a table.

For us it was clearer to present the results like we did it, the main reason why we added the table in the supplementary material being that the article can be read at the same time with the supplementary. Also, can be difficult to mention the personal explanations or possible implications in the table. Maybe it was confusing because that you did not have access to the supplementary material.

Q8. Remove Line 620

We removed it, thank you.

Reviewer 2 Report

This is an retrospective single center study on the management of sleep apnea patients during COVID pandemic. Although the study population is not large, the results gained are of scientific merit and interest for the filed of sleep medicine.

Minor issues: 

- Title should be changes in such manner to put ":" instead of "  , "

- Introduction section should briefly summarize the picks of pandemics in the investigated period in the current country

-In methodology section, clearly state exclusion criteria (not just as negation of inclusion criteria, enlist them as you did for inclusion)

-Discussion would benefit if elaborating the effects of sleep fragmentation and its mechanisms on anxiety (e.g. PMID: 34471462 and 31269081) is done, what is suggested. 

- Minor spelling errors should be ameliorated throughout the text 

Author Response

Thank you very much for your review, which I find very useful in significantly improving our article. We have provided point-by-point answers to your observations below.

Q1. Title should be changes in such manner to put ":" instead of "  , "

Thank you for the suggestion, we modified the title.

Q2. Introduction section should briefly summarize the picks of pandemics in the investigated period in the current country

In the introduction section, we wanted to summarize the main problems debated in the article, not only the local ones and to explain the data used to write the article, but the ideas also which were used to elaborate the study and after this, the article. In addition, we wanted to point out the liaison between OSA data, pandemic influence and mental health, reason why we expanded that much the introduction section.

Also, the evolution of the pandemic was the same in Romania as in other developed countries, that being the reason we used this data.

Q3. In methodology section, clearly state exclusion criteria (not just as negation of inclusion criteria, enlist them as you did for inclusion)

Thank you for the suggestion, we added the exclusion criteria (Lines198-201).

Q4. Discussion would benefit if elaborating the effects of sleep fragmentation and its mechanisms on anxiety (e.g. PMID: 34471462 and 31269081) is done, what is suggested.

We found very interesting the articles and, we included them in the discussion section. Indeed, there is an interesting connection between the sleep fragmentation and anxiety level explained through oxidative stress and hormonal changes. Thank you for the suggestion! (Reference 74 and 75 and lines 565-568)

Q5. Minor spelling errors should be ameliorated throughout the text.

We modified them as you suggested.

Reviewer 3 Report

The management of obstructive sleep apnea patients during the COVID-19 pandemic, a public health problem - interactions with sleep efficacy and mental health

Overview:

The paper reports on an experiment that evaluated how patients managed sleep apnea during the COVID-19 pandemic including CPAP usage changes and how these management changes were modified by a number of individual characteristics such as stress

General/Larger Issues:

There are some grammatical and other language issues. Recommend English language editor. I have started to go through it but with the many minor language problems it is difficult to determine exactly what is meant.

Section Specific Issues:

Title:

1.      I would suggest shortening the title from: The management of obstructive sleep apnea patients during the COVID-19 pandemic, a public health problem – interactions with sleep efficacy and mental health (22 words)

To something like: Sleep efficacy and mental health management of OSA during COVID

Abstract:

1.       rewrite with English language edit, sentences are comprehensible but clunky. For instance, I have highlighted the problematic phrase below :The aim was to evaluate how patients managed with  sleep apnea during COVID-19, to determine if continuous positive airway pressure (CPAP) usage changed after the beginning of the pandemic, to compare the stress level with the prepandemic one  and to observe if its modifications are related to their individual characteristics. – it should read something like “to compare baseline stress levels to those during the pandemic…”.

2.      Also watch tense (corrected instance above).

Introduction:

1.       Missing refs for first few claims of introduction

2.       OSA definition is not correct, I think it may just be an and instead of an or but not sure

Methods:

1.      Justify sample size

2.      Language issues again

3.      Need to ref Declaration of Helsinki

4.      The “perceived” stress questionnaire is not really an objective measure of stress, that would be something like HRV or cortisol levels

Other Minor Issues:

                                                              i.      The manuscript would also be more readable if the unnecessary acronyms were removed and their usage made consistent

                                                            ii.      Ln 22,25 – Tense problems

                                                          iii.      Ln 31 and many others – too many unnecessary “the” used

                                                          iv.      Line 26, 29 – use p< x rather than p = 0.00 – this is not a meaningful expression

Author Response

Thank you very much for your review, which I find very useful in significantly improving our article. We have provided point-by-point answers to your observations below.

General/Larger Issues: There are some grammatical and other language issues. Recommend English language editor. I have started to go through it but with the many minor language problems it is difficult to determine exactly what is meant.

We have put all the efforts to improve our language editing taking into consideration the Cambridge certificate of our authors.

Section Specific Issues:

Title:

  1. I would suggest shortening the title from: The management of obstructive sleep apnea patients during the COVID-19 pandemic, a public health problem – interactions with sleep efficacy and mental health (22 words). To something like: Sleep efficacy and mental health management of OSA during COVID

 Thank you for the suggestion. It was hard to find a title that include all the analyzed data, considering that we also included clinical data and other measurements, not only the parameters regarding the sleep efficacy. In addition, we were not limited to a specific length, reason why we wanted to be clearer in the title in order to make the article of interest for the readers.

Abstract:

  1. rewrite with English language edit, sentences are comprehensible but clunky. For instance, I have highlighted the problematic phrase below :The aim was to evaluate how patients managed with sleep apnea during COVID-19, to determine if continuous positive airway pressure (CPAP) usage changed after the beginning of the pandemic, to compare the stress level with the prepandemic one  and to observe if its modifications are related to their individual characteristics. – it should read something like “to compare baseline stress levels to those during the pandemic…”.

 Thank you very much for the recommendations. We modified the abstract (lines 20-34).

  1. Also watch tense (corrected instance above).

We corrected the abstract as you suggested (lines 20-34).

Introduction:

  1. Missing refs for first few claims of introduction

The information from the first few claims of the introduction is retrieved in the 1-10 refs, as specified in the brackets (lines 40,46,48,57).

  1. OSA definition is not correct, I think it may just be an and instead of an or but not sure

 Thank you for the suggestion, we checked again and updated the information based on the refs (line 63).

Methods:

  1. Justify sample size

The small sample size is mentioned to be a limitation and as is explained our sleep laboratory was closed due to the spread of COVID-19 (lines 611-615).

  1. Language issues again

We have put all the efforts to improve our language editing.

  1. Need to ref Declaration of Helsinki

We added the reference for the Declaration of Helsinki as you suggested (line 209, ref no. 48).

  1. The “perceived” stress questionnaire is not really an objective measure of stress, that would be something like HRV or cortisol levels

 As you mentioned, indeed, the PSQ is not really an objective measure of stress but is a classical instrument to evaluate the stress level. We modified in the test according to your suggestion (line 227).

Other Minor Issues:

  1. The manuscript would also be more readable if the unnecessary acronyms were removed and their usage made consistent.

Taking into account that the terms are repeating being the terminology from sleep medicine, it was a challenge to use as few acronyms as we could. We tried to reduce to the minimum the number of acronyms.

  1. Ln 22,25 – Tense problems

We corrected the abstract according to your suggestion (line 22-26).

  1. Ln 31 and many others – too many unnecessary “the” used

We corrected the abstract according to your suggestion.

  1. Line 26, 29 – use p< x rather than p = 0.00 – this is not a meaningful expression

 We corrected the information according to your suggestion, thank you (lines 27,31).

Round 2

Reviewer 3 Report

still numerous small errors and typos but much better, maybe also get someone to help with reviewer responses as they also didn't make a lot of sense at times

Author Response

Thank you very much for your review, which I find very useful in significantly improving our article.

  1. Still numerous small errors and typos but much better, maybe also get someone to help with reviewer responses as they also didn't make a lot of sense at times

Answer: Thank you for the suggestion, we have tried and have put all our efforts to improve the manuscript. We have highlighted the modifications with track changes. 
